# Semi-supervised Deep Kernel Learning: Regression with Unlabeled Data by Minimizing Predictive Variance

**Neal Jean,**[*] **Sang Michael Xie,**[*] **Stefano Ermon**
Department of Computer Science
Stanford University
Stanford, CA 94305
`{nealjean, xie, ermon}@cs.stanford.edu`

## Abstract

Large amounts of labeled data are typically required to train deep learning models. For many real-world problems, however, acquiring additional data can be expensive or even impossible. We present semi-supervised deep kernel learning (SSDKL), a semi-supervised regression model based on minimizing predictive variance in the posterior regularization framework. SSDKL combines the hierarchical representation learning of neural networks with the probabilistic modeling capabilities of Gaussian processes. By leveraging unlabeled data, we show improvements on a diverse set of real-world regression tasks over supervised deep kernel learning and semi-supervised methods such as VAT and mean teacher adapted for regression.

## 1 Introduction

The prevailing trend in machine learning is to automatically discover good feature representations through end-to-end optimization of neural networks. However, most success stories have been enabled by vast quantities of labeled data [1]. This need for supervision poses a major challenge when we encounter critical scientific and societal problems where fine-grained labels are difficult to obtain. Accurately measuring the outcomes that we care about—e.g., childhood mortality, environmental damage, or extreme poverty—can be prohibitively expensive [2, 3, 4]. Although these problems have limited data, they often contain underlying structure that can be used for learning; for example, poverty and other socioeconomic outcomes are strongly correlated over both space and time.

Semi-supervised learning approaches offer promise when few labels are available by allowing models to supplement their training with unlabeled data [5]. Mostly focusing on classification tasks, these methods often rely on strong assumptions about the structure of the data (e.g., cluster assumptions, low data density at decision boundaries [6]) that generally do not apply to regression [7, 8, 9, 10, 11].

In this paper, we present semi-supervised deep kernel learning, which addresses the challenge of semi-supervised regression by building on previous work combining the feature learning capabilities of deep neural networks with the ability of Gaussian processes to capture uncertainty [12, 3, 13]. SSDKL incorporates unlabeled training data by minimizing predictive variance in the posterior regularization framework, a flexible way of encoding prior knowledge in Bayesian models [14, 15, 16].

Our main contributions are the following:

- We introduce semi-supervised deep kernel learning (SSDKL) for the largely unexplored domain of deep semi-supervised regression. SSDKL is a regression model that combines

---

[*]denotes equal contribution

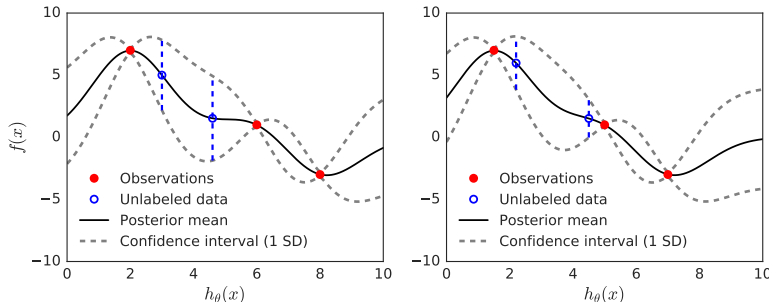

Figure 1: Depiction of the variance minimization approach behind semi-supervised deep kernel learning (SSDKL). The $x$-axis represents one dimension of a neural network embedding and the $y$-axis represents the corresponding output. **Left:** Without unlabeled data, the model learns an embedding by maximizing the likelihood of labeled data. The black and gray dotted lines show the posterior distribution after conditioning. **Right:** Embedding learned by SSDKL tries to minimize the predictive variance of unlabeled data, encouraging unlabeled embeddings to be near labeled embeddings. Observe that the representations of both labeled and unlabeled data are free to change.

the strengths of heavily parameterized deep neural networks and nonparametric Gaussian processes. While the deep Gaussian process kernel induces structure in an embedding space, the model also allows *a priori* knowledge of structure (i.e., spatial or temporal) in the input features to be naturally incorporated through kernel composition.

- By formalizing the semi-supervised variance minimization objective in the posterior regularization framework, we unify previous semi-supervised approaches such as minimum entropy and minimum variance regularization under a common framework. To our knowledge, this is the first paper connecting semi-supervised methods to posterior regularization.

- We demonstrate that SSDKL can use unlabeled data to learn more generalizable features and improve performance on a range of regression tasks, outperforming the supervised deep kernel learning method and semi-supervised methods such as virtual adversarial training (VAT) and mean teacher [17, 18]. In a challenging real-world task of predicting poverty from satellite images, SSDKL outperforms the state-of-the-art by $15.5\%$—by incorporating prior knowledge of spatial structure, the median improvement increases to $17.9\%$.

## 2   Preliminaries

We assume a training set of $n$ labeled examples $\{(\mathbf{x}_i, y_i)\}_{i=1}^n$ and $m$ unlabeled examples $\{\mathbf{x}_j\}_{j=n+1}^{n+m}$ with instances $\mathbf{x} \in \mathbb{R}^d$ and labels $y \in \mathbb{R}$. Let $X_L, \mathbf{y}_L, X_U$ refer to the aggregated features and targets, where $X_L \in \mathbb{R}^{n \times d}$, $\mathbf{y}_L \in \mathbb{R}^n$, and $X_U \in \mathbb{R}^{m \times d}$. At test time, we are given examples $X_T \in \mathbb{R}^{t \times d}$ that we would like to predict.

There are two major paradigms in semi-supervised learning, inductive and transductive. In *inductive* semi-supervised learning, the labeled data $(X_L, \mathbf{y}_L)$ and unlabeled data $X_U$ are used to learn a function $f : \mathcal{X} \mapsto \mathcal{Y}$ that generalizes well and is a good predictor on unseen test examples $X_T$ [5]. In *transductive* semi-supervised learning, the unlabeled examples are exactly the test data that we would like to predict, i.e., $X_T = X_U$ [19]. A transductive learning approach tries to find a function $f : \mathcal{X}^{n+m} \mapsto \mathcal{Y}^{n+m}$, with no requirement of generalizing to additional test examples. Although the theoretical development of SSDKL is general to both the inductive and transductive regimes, we only test SSDKL in the inductive setting in our experiments for direct comparison against supervised learning methods.

**Gaussian processes**   A Gaussian process (GP) is a collection of random variables, any finite number of which form a multivariate Gaussian distribution. Following the notation of [20], a Gaussian process defines a distribution over functions $f : \mathbb{R}^d \to \mathbb{R}$ from inputs to target values. If

$$f(\mathbf{x}) \sim \mathcal{GP}\left(\mu(\mathbf{x}), k_\phi(\mathbf{x}_i, \mathbf{x}_j)\right)$$

with mean function $\mu(\mathbf{x})$ and covariance kernel function $k_\phi(\mathbf{x}_i, \mathbf{x}_j)$ parameterized by $\phi$, then any collection of function values is jointly Gaussian,

$$f(X) = [f(\mathbf{x}_1), \ldots, f(\mathbf{x}_n)]^T \sim \mathcal{N}(\boldsymbol{\mu}, K_{X,X}),$$

with mean vector and covariance matrix defined by the GP, s.t. $\boldsymbol{\mu}_i = \mu(\mathbf{x}_i)$ and $(K_{X,X})_{ij} = k_\phi(\mathbf{x}_i, \mathbf{x}_j)$. In practice, we often assume that observations include i.i.d. Gaussian noise, i.e., $y(\mathbf{x}) = f(\mathbf{x}) + \epsilon(\mathbf{x})$ where $\epsilon \sim \mathcal{N}(0, \phi_n^2)$, and the covariance function becomes

$$\text{Cov}(y(\mathbf{x}_i), y(\mathbf{x}_j)) = k(\mathbf{x}_i, \mathbf{x}_j) + \phi_n^2 \delta_{ij}$$

where $\delta_{ij} = \mathbb{I}[i = j]$. To make predictions at unlabeled points $X_U$, we can compute a Gaussian posterior distribution in closed form by conditioning on the observed data $(X_L, \mathbf{y}_L)$. For a more thorough introduction, we refer readers to [21].

**Deep kernel learning**    Deep kernel learning (DKL) combines neural networks with GPs by using a neural network embedding as input to a deep kernel [12]. Given input data $\mathbf{x} \in \mathcal{X}$, a neural network parameterized by $w$ is used to extract features $h_w(\mathbf{x}) \in \mathbb{R}^p$. The outputs are modeled as

$$f(\mathbf{x}) \sim \mathcal{GP}(\mu(h_w(\mathbf{x})), k_\phi(h_w(\mathbf{x}_i), h_w(\mathbf{x}_j)))$$

for some mean function $\mu(\cdot)$ and base kernel function $k_\phi(\cdot, \cdot)$ with parameters $\phi$. Parameters $\theta = (w, \phi)$ of the deep kernel are learned jointly by minimizing the negative log likelihood of the labeled data [20]:

$$L_{likelihood}(\theta) = -\log p(\mathbf{y}_L \mid X_L, \theta) \tag{1}$$

For Gaussian distributions, the marginal likelihood is a closed-form, differentiable expression, allowing DKL models to be trained via backpropagation.

**Posterior regularization**    In probabilistic models, domain knowledge is generally imposed through the specification of priors. These priors, along with the observed data, determine the posterior distribution through the application of Bayes' rule. However, it can be difficult to encode our knowledge in a Bayesian prior. *Posterior regularization* offers a more direct and flexible mechanism for controlling the posterior distribution.

Let $\mathcal{D} = (X_L, \mathbf{y}_L)$ be a collection of observed data. [15] present a regularized optimization formulation called *regularized Bayesian inference*, or RegBayes. In this framework, the regularized posterior is the solution of the following optimization problem:

$$\textbf{RegBayes:} \quad \inf_{q(M|\mathcal{D}) \in \mathcal{P}_{prob}} \mathcal{L}(q(M|\mathcal{D})) + \Omega(q(M|\mathcal{D})) \tag{2}$$

where $\mathcal{L}(q(M|\mathcal{D}))$ is defined as the KL-divergence between the desired post-data posterior $q(M|\mathcal{D})$ over models $M$ and the standard Bayesian posterior $p(M|\mathcal{D})$ and $\Omega(q(M|\mathcal{D}))$ is a posterior regularizer. The goal is to learn a posterior distribution that is not too far from the standard Bayesian posterior while also fulfilling some requirements imposed by the regularization.

## 3   Semi-supervised deep kernel learning

We introduce *semi-supervised deep kernel learning* (SSDKL) for problems where labeled data is limited but unlabeled data is plentiful. To learn from unlabeled data, we observe that a Bayesian approach provides us with a predictive posterior *distribution*—i.e., we are able to quantify predictive uncertainty. Thus, we regularize the posterior by adding an unsupervised loss term that minimizes the predictive variance at unlabeled data points:

$$L_{semisup}(\theta) = \frac{1}{n} L_{likelihood}(\theta) + \frac{\alpha}{m} L_{variance}(\theta) \tag{3}$$

$$L_{variance}(\theta) = \sum_{x \in X_U} \text{Var}(f(x)) \tag{4}$$

where $n$ and $m$ are the numbers of labeled and unlabeled training examples, $\alpha$ is a hyperparameter controlling the trade-off between supervised and unsupervised components, and $\theta$ represents the model parameters.

## 3.1 Variance minimization as posterior regularization

Optimizing $L_{semisup}$ is equivalent to computing a regularized posterior through solving a specific instance of the RegBayes optimization problem (2), where our choice of regularizer corresponds to variance minimization.

Let $X = (X_L, X_U)$ be the observed input data and $\mathcal{D} = (X, \mathbf{y}_L)$ be the input data with labels for the labeled part $X_L$. Let $\mathcal{F}$ denote a space of functions where for $f \in \mathcal{F}$, $f : \mathbb{R}^d \to \mathbb{R}$ maps from the inputs to the target values. Note that here, $M = (f, \theta)$ is the model in the RegBayes framework, where $\theta$ are the model parameters. We assume that the prior is $\pi(f, \theta)$ and a likelihood density $p(\mathcal{D}|f, \theta)$ exists. Given observed data $\mathcal{D}$, the Bayesian posterior is $p(f, \theta|\mathcal{D})$, while RegBayes computes a different, regularized posterior.

Let $\bar{\theta}$ be a specific instance of the model parameters. Instead of maximizing the marginal likelihood of the labeled training data in a purely supervised approach, we train our model in a semi-supervised fashion by minimizing the compound objective

$$L_{semisup}(\bar{\theta}) = -\frac{1}{n}\log p(\mathbf{y}_L|X_L, \bar{\theta}) + \frac{\alpha}{m}\sum_{x \in X_U}\mathrm{Var}_{f \sim p}(f(x)) \qquad (5)$$

where the variance is with respect to $p(f|\bar{\theta}, \mathcal{D})$, the Bayesian posterior given $\bar{\theta}$ and $\mathcal{D}$.

**Theorem 1.** *Let observed data $\mathcal{D}$, a suitable space of functions $\mathcal{F}$, and parameter space $\Theta$ be given. As in [15], we assume that $\mathcal{F}$ is a complete separable metric space and $\Pi$ is an absolutely continuous probability measure (with respect to background measure $\eta$) on $(\mathcal{F}, \mathcal{B}(\mathcal{F}))$, where $\mathcal{B}(\mathcal{F})$ is the Borel $\sigma$-algebra, such that a density $\pi$ exists where $d\Pi = \pi d\eta$ and we have prior density $\pi(f, \theta)$ and likelihood density $p(\mathcal{D}|f, \theta)$. Then the semi-supervised variance minimization problem (5)*

$$\inf_{\bar{\theta}} L_{semisup}(\bar{\theta})$$

*is equivalent to the RegBayes optimization problem (2)*

$$\inf_{q(f, \theta|\mathcal{D}) \in \mathcal{P}_{prob}} \mathcal{L}(q(f, \theta|\mathcal{D})) + \Omega(q(f, \theta|\mathcal{D}))$$

$$\Omega(q(f, \theta|\mathcal{D})) = \alpha' \sum_{i=1}^{m}\left( \int_{f, \theta} p(f|\theta, \mathcal{D})q(\theta|\mathcal{D})(f(X_U)_i - \mathbb{E}_p[f(X_U)_i])^2 d\eta(f, \theta)\right),$$

*where $\alpha' = \frac{\alpha n}{m}$, and $\mathcal{P}_{prob} = \{q : q(f, \theta|\mathcal{D}) = q(f|\theta, \mathcal{D})\delta_{\bar{\theta}}(\theta|\mathcal{D}), \bar{\theta} \in \Theta\}$ is a variational family of distributions where $q(\theta|\mathcal{D})$ is restricted to be a Dirac delta centered on $\bar{\theta} \in \Theta$.*

We include a formal derivation in Appendix A.1 and give a brief outline here. It can be shown that solving the variational optimization objective

$$\inf_{q(f, \theta|\mathcal{D})} D_{KL}(q(f, \theta|\mathcal{D})\|\pi(f, \theta)) - \int_{f, \theta} q(f, \theta|\mathcal{D}) \log p(\mathcal{D}|f, \theta) d\eta(f, \theta) \qquad (6)$$

is equivalent to minimizing the unconstrained form of the first term $\mathcal{L}(q(f, \theta|\mathcal{D}))$ of the RegBayes objective in Theorem 1, and the minimizer is precisely the Bayesian posterior $p(f, \theta|\mathcal{D})$. When we restrict the optimization to $q \in \mathcal{P}_{prob}$ the solution is of the form $q^*(f, \theta|\mathcal{D}) = p(f|\theta, \mathcal{D})\delta_{\bar{\theta}}(\theta|\mathcal{D})$ for some $\bar{\theta}$. This allows us to show that (6) is also equivalent to minimizing the first term of $L_{semisup}(\bar{\theta})$. Finally, noting that the regularization function $\Omega$ only depends on $\bar{\theta}$ (through $q(\theta|\mathcal{D}) = \delta_{\bar{\theta}}(\theta)$), the form of $q^*(f, \theta|\mathcal{D})$ is unchanged after adding $\Omega$. Therefore the choice of $\Omega$ reduces to minimizing the predictive variance with respect to $q^*(f|\theta, \mathcal{D}) = p(f|\bar{\theta}, \mathcal{D})$.

**Intuition for variance minimization** By minimizing $L_{semisup}$, we trade off maximizing the likelihood of our observations with minimizing the posterior variance on unlabeled data that we wish to predict. The posterior variance acts as a proxy for distance with respect to the kernel function in the deep feature space, and the regularizer is an inductive bias on the structure of the feature space. Since the deep kernel parameters are jointly learned, the neural net is encouraged to learn a feature representation in which the unlabeled examples are closer to the labeled examples, thereby reducing the variance on our predictions. If we imagine the labeled data as "supports" for the

surface representing the posterior mean, we are optimizing for embeddings where unlabeled data tend to cluster around these labeled supports. In contrast, the variance regularizer would not benefit conventional GP learning since fixed kernels would not allow for adapting the relative distances between data points.

Another interpretation is that the semi-supervised objective is a regularizer that reduces overfitting to labeled data. The model is discouraged from learning features from labeled data that are not also useful for making low-variance predictions at unlabeled data points. In settings where unlabeled data provide additional variation beyond labeled examples, this can improve model generalization.

**Training and inference**    Semi-supervised deep kernel learning scales well with large amounts of unlabeled data since the unsupervised objective $L_{variance}$ naturally decomposes into a sum over conditionally independent terms. This allows for mini-batch training on *unlabeled* data with stochastic gradient descent. Since all of the labeled examples are interdependent, computing exact gradients for *labeled* examples requires full batch gradient descent on the labeled data. Therefore, assuming a constant batch size, each iteration of training requires $O(n^3)$ computations for a Cholesky decomposition, where $n$ is the number of labeled training examples. Performing the GP inference requires $O(n^3)$ one-time cost in the labeled points. However, existing approximation methods based on kernel interpolation and structured matrices used in DKL can be directly incorporated in SSDKL and would reduce the training complexity to close to linear in labeled dataset size and inference to constant time per test point [12, 22]. While DKL is designed for the supervised setting where scaling to large labeled datasets is a very practical concern, our focus is on semi-supervised settings where labels are limited but unlabeled data is abundant.

# 4    Experiments and results

We apply SSDKL to a variety of real-world regression tasks in the inductive semi-supervised learning setting, beginning with eight datasets from the UCI repository [23]. We also explore the challenging task of predicting local poverty measures from high-resolution satellite imagery [24]. In our reported results, we use the squared exponential or radial basis function kernel. We also experimented with polynomial kernels, but saw generally worse performance. Our SSDKL model is implemented in TensorFlow [25]. Additional training details are provided in Appendix A.3, and code and data for reproducing experimental results can be found on GitHub.[2]

## 4.1    Baselines

We first compare SSDKL to the purely supervised DKL, showing the contribution of unlabeled data. In addition to the supervised DKL method, we compare against semi-supervised methods including co-training, consistency regularization, generative modeling, and label propagation. Many of these methods were originally developed for semi-supervised classification, so we adapt them here for regression. All models, including SSDKL, were trained from random initializations.

COREG, or CO-training REGressors, uses two $k$-nearest neighbor ($k$NN) regressors, each of which generates labels for the other during the learning process [26]. Unlike traditional co-training, which requires splitting features into sufficient and redundant views, COREG achieves regressor diversity by using different distance metrics for its two regressors [27].

Consistency regularization methods aim to make model outputs invariant to local input perturbations [17, 28, 18]. For semi-supervised classification, [29] found that VAT and mean teacher were the best methods using fair evaluation guidelines. Virtual adversarial training (VAT) via local distributional smoothing (LDS) enforces consistency by training models to be robust to adversarial local input perturbations [17, 30]. Unlike adversarial training [31], the *virtual* adversarial perturbation is found without labels, making semi-supervised learning possible. We adapt VAT for regression by choosing the output distribution $\mathcal{N}(h_\theta(\mathbf{x}), \sigma^2)$ for input $\mathbf{x}$, where $h_\theta : \mathbb{R}^d \to \mathbb{R}$ is a parameterized mapping and $\sigma$ is fixed. Optimizing the likelihood term is then equivalent to minimizing squared error; the LDS term is the KL-divergence between the model distribution and a perturbed Gaussian (see Appendix A.2). Mean teacher enforces consistency by penalizing deviation from the outputs of a model with the exponential weighted average of the parameters over SGD iterations [18].

| | | | Percent reduction in RMSE compared to DKL | | | | | | | | | | | |
|---|---|---|---|---|---|---|---|---|---|---|---|---|---|---|
| | | | $n=100$ | | | | | | $n=300$ | | | | | |
| Dataset | $N$ | $d$ | SSDKL | COREG | Label Prop | VAE | Mean Teacher | VAT | SSDKL | COREG | Label Prop | VAE | Mean Teacher | VAT |
| Skillcraft | 3,325 | 18 | 3.44 | 1.87 | **5.12** | 0.11 | -19.72 | -21.97 | **5.97** | 0.60 | 5.78 | 4.36 | -18.17 | -20.13 |
| Parkinsons | 5,875 | 20 | **-2.51** | -27.45 | -43.43 | -122.23 | -91.54 | -143.60 | **5.97** | -22.50 | -51.35 | -167.93 | -132.68 | -202.79 |
| Elevators | 16,599 | 18 | **7.99** | -5.22 | 2.28 | -22.68 | -27.27 | -31.25 | **6.92** | -25.98 | -22.08 | -53.40 | -82.01 | -63.68 |
| Protein | 45,730 | 9 | -3.34 | -2.37 | **0.77** | -8.65 | -5.11 | -6.44 | 1.23 | 0.89 | **2.61** | -9.24 | -8.98 | -10.38 |
| Blog | 52,397 | 280 | 5.65 | **11.15** | 9.01 | 8.96 | 7.05 | 1.87 | 5.34 | 9.60 | **12.44** | 8.14 | 7.87 | 9.08 |
| CTslice | 53,500 | 384 | -22.48 | **-12.11** | -17.12 | -47.59 | -60.71 | -64.75 | **5.64** | 2.94 | -2.59 | -60.18 | -58.97 | -84.60 |
| Buzz | 583,250 | 77 | 5.59 | **13.80** | 1.33 | -19.26 | -77.08 | -82.66 | **11.33** | 10.41 | -2.22 | -28.65 | -104.88 | -100.82 |
| Electric | 2,049,280 | 6 | **4.96** | -126.95 | -201.18 | -285.61 | -399.85 | -513.95 | **-13.93** | -154.07 | -303.21 | -460.48 | -627.83 | -828.35 |
| Median | | | **4.20** | -3.80 | 1.05 | -20.97 | -43.99 | -48.00 | **5.81** | 0.75 | -2.41 | -41.02 | -70.49 | -74.14 |

Table 1: Percent reduction in RMSE compared to baseline supervised deep kernel learning (DKL) model for semi-supervised deep kernel learning (SSDKL), COREG, label propagation, variational auto-encoder (VAE), mean teacher, and virtual adversarial training (VAT) models. Results are averaged across 10 trials for each UCI regression dataset. Here $N$ is the total number of examples, $d$ is the input feature dimension, and $n$ is the number of labeled training examples. Final row shows median percent reduction in RMSE achieved by using unlabeled data.

Label propagation defines a graph structure over the data with edges that define the probability for a categorical label to propagate from one data point to another [32]. If we encode this graph in a transition matrix $T$ and let the current class probabilities be $y$, then the algorithm iteratively propagates $y \leftarrow Ty$, row-normalizes $y$, clamps the labeled data to their known values, and repeats until convergence. We make the extension to regression by letting $y$ be real-valued labels and normalizing $T$. As in [32], we use a fully-connected graph and the radial-basis kernel for edge weights. The kernel scale hyperparameter is chosen using a validation set.

Generative models such as the variational autoencoder (VAE) have shown promise in semi-supervised classification especially for visual and sequential tasks [33, 34, 35, 36]. We compare against a semi-supervised VAE by first learning an unsupervised embedding of the data and then using the embeddings as input to a supervised multilayer perceptron.

## 4.2 UCI regression experiments

We evaluate SSDKL on eight regression datasets from the UCI repository. For each dataset, we train on $n = \{50, 100, 200, 300, 400, 500\}$ labeled examples, retain 1000 examples as the hold out test set, and treat the remaining data as unlabeled examples. Following [29], the labeled data is randomly split 90-10 into training and validation samples, giving a realistically small validation set. For example, for $n = 100$ labeled examples, we use 90 random examples for training and the remaining 10 for validation in every random split. We report test RMSE averaged over 10 trials of random splits to combat the small data sizes. All kernel hyperparameters are optimized directly through $L_{semisup}$, and we use the validation set for early stopping to prevent overfitting and for selecting $\alpha \in \{0.1, 1, 10\}$. We did not use approximate GP procedures in our SSDKL or DKL experiments, so the only difference is the addition of the variance regularizer. For all combinations of input feature dimensions and labeled data sizes in the UCI experiments, each SSDKL trial (including all training and testing) ran on the order of minutes.

Following [20], we choose a neural network with a similar [$d$-100-50-50-2] architecture and two-dimensional embedding. Following [29], we use this same base model for all deep models, including SSDKL, DKL, VAT, mean teacher, and the VAE encoder, in order to make results comparable across methods. Since label propagation creates a kernel matrix of all data points, we limit the number of unlabeled examples for label propagation to a maximum of 20000 due to memory constraints. We initialize labels in label propagation with a kNN regressor with $k = 5$ to speed up convergence.

Table 1 displays the results for $n = 100$ and $n = 300$; full results are included in Appendix A.3. SSDKL gives a $4.20\%$ and $5.81\%$ median RMSE improvement over the supervised DKL in the $n = 100, 300$ cases respectively, superior to other semi-supervised methods adapted for regression. A Wilcoxon signed-rank test versus DKL shows significance at the $p = 0.05$ level for at least one labeled training set size for all 8 datasets.

The same learning rates and initializations are used across *all* UCI datasets for SSDKL. We use learning rates of $1 \times 10^{-3}$ and 0.1 for the neural network and GP parameters respectively and

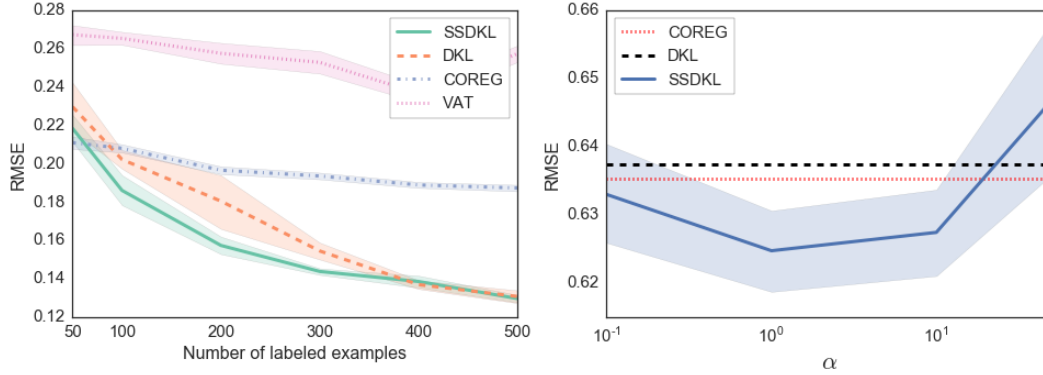

Figure 2: **Left:** Average test RMSE vs. number of labeled examples for UCI Elevators dataset, $n = \{50, 100, 200, 300, 400, 500\}$. SSDKL generally outperforms supervised DKL, co-training regressors (COREG), and virtual adversarial training (VAT). **Right:** SSDKL performance on poverty prediction (Section 4.3) as a function of $\alpha$, which controls the trade-off between labeled and unlabeled objectives, for $n = 300$. The dotted lines plot the performance of DKL and COREG. All results averaged over 10 trials. In both panels, shading represents one standard deviation.

initialize all GP parameters to 1. In Fig. 2 (**right**), we study the effect of varying $\alpha$ to trade off between maximizing the likelihood of labeled data and minimizing the variance of unlabeled data. A large $\alpha$ emphasizes minimization of the predictive variance while a small $\alpha$ focuses on fitting labeled data. SSDKL improves on DKL for values of $\alpha$ between 0.1 and 10.0, indicating that performance is not overly reliant on the choice of this hyperparameter. Fig. 2 (**left**) compares SSDKL to purely supervised DKL, COREG, and VAT as we vary the labeled training set size. For the Elevators dataset, DKL is able to close the gap on SSDKL as it gains access to more labeled data. Relative to the other methods, which require more data to fit neural network parameters, COREG performs well in the low-data regime.

Surprisingly, COREG outperformed SSDKL on the Blog, CTslice, and Buzz datasets. We found that these datasets happen to be better-suited for nearest neighbors-based methods such as COREG. A kNN regressor using only the labeled data outperformed DKL on two of three datasets for $n = 100$, beat SSDKL on all three for $n = 100$, beat DKL on two of three for $n = 300$, and beat SSDKL on one of three for $n = 300$. Thus, the kNN regressor is often already outperforming SSDKL with only labeled data—it is unsurprising that SSDKL is unable to close the gap on a semi-supervised nearest neighbors method like COREG.

**Representation learning**    To gain some intuition about how the unlabeled data helps in the learning process, we visualize the neural network embeddings learned by the DKL and SSDKL models on the Skillcraft dataset. In Fig. 3 (**left**), we first train DKL on $n = 100$ labeled training examples and plot the two-dimensional neural network embedding that is learned. In Fig. 3 (**right**), we train SSDKL on $n = 100$ labeled training examples along with $m = 1000$ additional unlabeled data points and plot the resulting embedding. In the left panel, DKL learns a poor embedding—different colors representing different output magnitudes are intermingled. In the right panel, SSDKL is able to use the unlabeled data for regularization, and learns a better representation of the dataset.

## 4.3   Poverty prediction

High-resolution satellite imagery offers the potential for cheap, scalable, and accurate tracking of changing socioeconomic indicators. In this task, we predict local poverty measures from satellite images using limited amounts of poverty labels. As described in [2], the dataset consists of $3,066$ villages across five Africa countries: Nigeria, Tanzania, Uganda, Malawi, and Rwanda. These include some of the poorest countries in the world (Malawi and Rwanda) as well as some that are relatively better off (Nigeria), making for a challenging and realistically diverse problem.

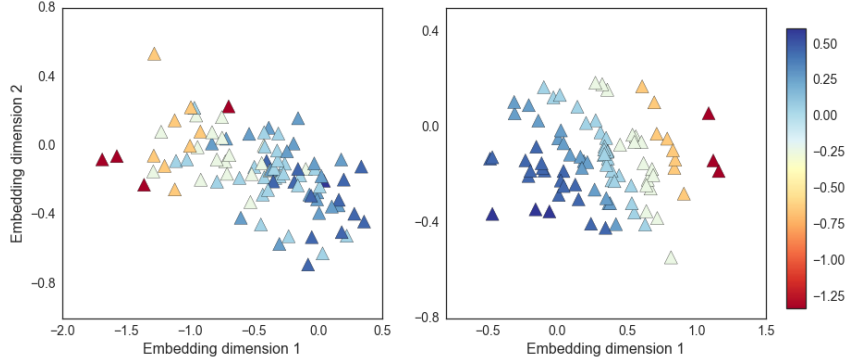

Figure 3: **Left:** Two-dimensional embeddings learned by supervised deep kernel learning (DKL) model on the Skillcraft dataset using $n = 100$ labeled training examples. The colorbar shows the magnitude of the normalized outputs. **Right:** Embeddings learned by semi-supervised deep kernel learning (SSDKL) model using $n = 100$ labeled examples plus an additional $m = 1000$ unlabeled examples. By using unlabeled data for regularization, SSDKL learns a better representation.

| | Percent reduction in RMSE ($n = 300$) | | |
|---|---|---|---|
| Country | Spatial SSDKL | SSDKL | DKL |
| Malawi | 13.7 | **16.4** | 15.7 |
| Nigeria | **17.9** | 4.6 | 1.7 |
| Tanzania | 10.0 | **15.5** | 9.2 |
| Uganda | **25.2** | 12.1 | 13.8 |
| Rwanda | **27.0** | 25.4 | 21.3 |
| Median | **17.9** | 15.5 | 13.8 |

Table 2: Percent RMSE reduction in a poverty measure prediction task compared to baseline ridge regression model used in [2]. SSDKL and DKL models use only satellite image data. Spatial SSDKL incorporates both location and image data through kernel composition. Final row shows median RMSE reduction of each model averaged over 10 trials.

In this experiment, we use $n = 300$ labeled satellite images for training. With such a small dataset, we can not expect to train a deep convolutional neural network (CNN) from scratch. Instead we take a transfer learning approach as in [24], extracting 4096-dimensional visual features and using these as input. More details are provided in Appendix A.5.

**Incorporating spatial information** In order to highlight the usefulness of kernel composition, we explore extending SSDKL with a spatial kernel. Spatial SSDKL composes two kernels by summing an image feature kernel and a separate location kernel that operates on location coordinates (lat/lon). By treating them separately, it explicitly encodes the knowledge that location coordinates are spatially structured and distinct from image features.

As shown in Table 2, all models outperform the baseline state-of-the-art ridge regression model from [2]. Spatial SSDKL significantly outperforms the DKL and SSDKL models that use only image features. Spatial SSDKL outperforms the other models by directly modeling location coordinates as spatial features, showing that kernel composition can effectively incorporate prior knowledge of structure.

# 5 Related work

[37] introduced deep Gaussian processes, which stack GPs in a hierarchy by modeling the outputs of one layer with a Gaussian process in the next layer. Despite the suggestive name, these models do not integrate deep neural networks and Gaussian processes.

[12] proposed deep kernel learning, combining neural networks with the non-parametric flexibility of GPs and training end-to-end in a fully supervised setting. Extensions have explored approximate inference, stochastic gradient training, and recurrent deep kernels for sequential data [22, 38, 39].

Our method draws inspiration from transductive experimental design, which chooses the most informative points (experiments) to measure by seeking data points that are both hard to predict and informative for the unexplored test data [40]. Similar prediction uncertainty approaches have been explored in semi-supervised classification models, such as minimum entropy and minimum variance regularization, which can now also be understood in the posterior regularization framework [7, 41].

Recent work in generative adversarial networks (GANs) [33], variational autoencoders (VAEs) [34], and other generative models have achieved promising results on various semi-supervised classification tasks [35, 36]. However, we find that these models are not as well-suited for generic regression tasks such as in the UCI repository as for audio-visual tasks.

Consistency regularization posits that the model's output should be invariant to reasonable perturbations of the input [17, 28, 18]. Combining adversarial training [31] with consistency regularization, virtual adversarial training uses a label-free regularization term that allows for semi-supervised training [17]. Mean teacher adds a regularization term that penalizes deviation from a exponential weighted average of the parameters over SGD iterations [18]. For semi-supervised classification, [29] found that VAT and mean teacher were the best methods across a series of fair evaluations.

Label propagation defines a graph structure over the data points and propagates labels from labeled data over the graph. The method must assume a graph structure and edge distances on the input feature space without the ability to adapt the space to the assumptions. Label propagation is also subject to memory constraints since it forms a kernel matrix over *all* data points, requiring quadratic space in general, although sparser graph structures can reduce this to a linear scaling.

Co-training regressors trains two kNN regressors with different distance metrics that label each others' unlabeled data. This works when neighbors in the given input space have similar target distributions, but unlike kernel learning approaches, the features are fixed. Thus, COREG cannot adapt the space to a misspecified distance measure. In addition, as a fully nonparametric method, inference requires retaining the full dataset.

Much of the previous work in semi-supervised learning is in classification and the assumptions do not generally translate to regression. Our experiments show that SSDKL outperforms other adapted semi-supervised methods in a battery of regression tasks.

## 6    Conclusions

Many important problems are challenging because of the limited availability of training data, making the ability to learn from unlabeled data critical. In experiments with UCI datasets and a real-world poverty prediction task, we find that minimizing posterior variance can be an effective way to incorporate unlabeled data when labeled training data is scarce. SSDKL models are naturally suited for many real-world problems, as spatial and temporal structure can be explicitly modeled through the composition of kernel functions. While our focus is on regression problems, we believe the SSDKL framework is equally applicable to classification problems—we leave this to future work.

**Acknowledgements**

This research was supported by NSF (#1651565, #1522054, #1733686), ONR, Sony, and FLI. NJ was supported by the Department of Defense (DoD) through the National Defense Science & Engineering Graduate Fellowship (NDSEG) Program. We are thankful to Volodymyr Kuleshov and Aditya Grover for helpful discussions.

## Footnotes

[2]https://github.com/ermongroup/ssdkl

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
