[Supplementary Material]

# Semi-supervised Deep Kernel Learning: Regression with Unlabeled Data by Minimizing Predictive Variance

**Neal Jean,**[*] **Sang Michael Xie,**[*] **Stefano Ermon**
Department of Computer Science
Stanford University
Stanford, CA 94305
{nealjean, xie, ermon}@cs.stanford.edu

## A   Appendix

### A.1   Posterior regularization

*Proof of Theorem 1.* Let $\mathcal{D} = (X_L, \mathbf{y}_L, X_U)$ be a collection of observed data. Let $X = (X_L, X_U)$ be the observed input data points. As in [1], we assume that $\mathcal{F}$ is a complete separable metric space and $\Pi$ is an absolutely continuous probability measure (with respect to background measure $\eta$) on $(\mathcal{F}, \mathcal{B}(\mathcal{F}))$, where $\mathcal{B}(\mathcal{F})$ is the the Borel $\sigma$-algebra, such that a density $\pi$ exists where $d\Pi = \pi d\eta$. Let $\Theta$ be a space of parameters to the model, where we treat $\theta \in \Theta$ as random variables. With regards to the notation in the RegBayes framework, the model is the pair $M = (f, \theta)$. We assume as in [1] that the likelihood function $P(\cdot|M)$ is the likelihood distribution which is dominated by a $\sigma$-finite measure $\lambda$ for all $M$ with positive density, such that a density $p(\cdot|M)$ exists where $dP(\cdot|M) = p(\cdot|M)d\lambda$.

We would like to compute the posterior distribution

$$p(f, \theta | \mathcal{D}) = \frac{p(\mathcal{D}|f, \theta)\pi(f, \theta)}{\int_{f,\theta} p(f, \theta, \mathcal{D})d\eta(f, \theta)}$$

which involves an intractable integral.

We claim that the solution of the following optimization problem is precisely the Bayesian posterior $p(f, \theta | \mathcal{D})$:

$$\inf_{q(f,\theta|\mathcal{D})} D_{KL}(q(f, \theta | \mathcal{D}) \| \pi(f, \theta)) - \int_{f,\theta} q(f, \theta | \mathcal{D}) \log p(\mathcal{D}|f, \theta)d\eta(f, \theta).$$

By adding the constant $\log p(\mathcal{D})$ to the objective,

$$\operatorname*{arginf}_{q(f,\theta|\mathcal{D})} D_{KL}(q(f, \theta | \mathcal{D}) \| \pi(f, \theta)) - \int_{f,\theta} q(f, \theta | \mathcal{D}) \log p(\mathcal{D}|f, \theta)d\eta(f, \theta) + \log p(\mathcal{D}) \tag{1}$$

$$= \operatorname*{arginf}_{q(f,\theta|\mathcal{D})} \int_{f,\theta} q(f, \theta | \mathcal{D}) \log \frac{q(f, \theta | \mathcal{D})}{p(f, \theta, \mathcal{D})} d\eta(f, \theta) + \log p(\mathcal{D}) \tag{2}$$

$$= \operatorname*{arginf}_{q(f,\theta|\mathcal{D})} \int_{f,\theta} q(f, \theta | \mathcal{D}) \log \frac{q(f, \theta | \mathcal{D})}{p(f, \theta | \mathcal{D})} d\eta(f, \theta) \tag{3}$$

$$= \operatorname*{arginf}_{q(f,\theta|\mathcal{D})} D_{KL}(q(f, \theta | \mathcal{D}) \| p(f, \theta | \mathcal{D})) \tag{4}$$

$$= \operatorname*{arginf}_{q(f,\theta|\mathcal{D})} \mathcal{L}(q(f, \theta | \mathcal{D})), \tag{5}$$

---

[*]denotes equal contribution

where by definition $p(f, \theta, \mathcal{D}) = p(\mathcal{D}|f, \theta)\pi(f, \theta)$ and we see that the objective is minimized when $q(f, \theta|\mathcal{D}) = p(f, \theta|\mathcal{D})$ as claimed. We note that the objective is equivalent to the first term of the RegBayes objective (Section 2.3).

We introduce a variational approximation $q \in \mathcal{P}_{prob}$ which approximates $p(f, \theta|\mathcal{D})$, where $\mathcal{P}_{prob} = \{q : q(f, \theta|\mathcal{D}) = q(f|\theta, \mathcal{D})\delta_{\bar{\theta}}(\theta|\mathcal{D})\}$ is the family of approximating distributions such that $q(\theta|\mathcal{D})$ is restricted to be a Dirac delta centered on $\bar{\theta}$. When we restrict $q \in \mathcal{P}_{prob}$,

$$\operatornamewithlimits{arginf}_{q(f,\theta|\mathcal{D})\in\mathcal{P}_{prob}} D_{KL}(q(f,\theta|\mathcal{D})\|\pi(f,\theta)) - \int_{f,\theta} q(f,\theta|\mathcal{D})\log p(\mathcal{D}|f,\theta)d\eta(f,\theta) \tag{6}$$

$$= \operatornamewithlimits{arginf}_{q(f|\theta,\mathcal{D}),\bar{\theta}} \int_{\theta} \delta_{\bar{\theta}}(\theta)\int_{f} q(f|\theta,\mathcal{D})\left(\log\frac{q(f|\theta,\mathcal{D})\delta_{\bar{\theta}}(\theta)}{p(f,\theta|\mathcal{D})}\right)d\eta(f,\theta) \tag{7}$$

$$= \operatornamewithlimits{arginf}_{q(f|\theta,\mathcal{D}),\bar{\theta}} \int_{\theta} \delta_{\bar{\theta}}(\theta)\left(D_{KL}(q(f|\theta,\mathcal{D})\|p(f|\theta,\mathcal{D})) + \log\delta_{\bar{\theta}}(\theta) - \log p(\theta|\mathcal{D})\right)d\eta(\theta) \tag{8}$$

$$= \operatornamewithlimits{arginf}_{q(f|\theta,\mathcal{D}),\bar{\theta}} \int_{\theta} \delta_{\bar{\theta}}(\theta)\left(D_{KL}(q(f|\theta,\mathcal{D})\|p(f|\theta,\mathcal{D})) - \log p(\theta|\mathcal{D})\right)d\eta(\theta) \tag{9}$$

where in equation (7) we use the form from equation (3), and in equation (9) we note that $\int_{\theta}\delta_{\bar{\theta}}(\theta)\log\delta_{\bar{\theta}}(\theta)$ does not vary with $\bar{\theta}$ or $q$, and can be removed from the optimization. For every $\theta$, the optimizing distribution is $q^*(f|\theta, \mathcal{D}) = p(f|\bar{\theta}, \mathcal{D})$, which is the Bayesian posterior given the model parameters. Substituting this optimal value into (9),

$$\operatornamewithlimits{arginf}_{q(f|\theta,\mathcal{D}),\bar{\theta}} \int_{\theta} \delta_{\bar{\theta}}(\theta)\left(D_{KL}(q^*(f|\theta,\mathcal{D})\|p(f|\theta,\mathcal{D})) - \log p(\theta|\mathcal{D})\right)d\eta(\theta) \tag{10}$$

$$= \operatornamewithlimits{arginf}_{\bar{\theta}} -\int_{\theta} \delta_{\bar{\theta}}(\theta)\log p(\theta|\mathcal{D})d\eta(\theta) \tag{11}$$

$$= \operatornamewithlimits{arginf}_{\bar{\theta}} -\int_{\theta} \delta_{\bar{\theta}}(\theta)(\log p(\theta|X,\mathbf{y}_L) + \log p(\mathbf{y}_L|X))d\eta(\theta) \tag{12}$$

$$= \operatornamewithlimits{arginf}_{\bar{\theta}} -\int_{\theta} \delta_{\bar{\theta}}(\theta)\log p(\mathbf{y}_L,\theta|X)d\eta(\theta) \tag{13}$$

$$= \operatornamewithlimits{arginf}_{\bar{\theta}} -\log p(\mathbf{y}_L|\bar{\theta},X) \tag{14}$$

using that $D_{KL}(q^*(f|\theta, \mathcal{D})\|p(f|\theta, \mathcal{D})) = 0$ in (11) and $\int_{\theta}\delta_{\bar{\theta}}(\theta)\log p(\mathbf{y}_L|X)$ is a constant. The optimization problem over $\bar{\theta}$ reflects maximizing the likelihood of the data.

We defined the regularization function as

$$\Omega(q(f,\theta|\mathcal{D})) = \alpha'\sum_{i=1}^{m}\left(\int_{f,\theta} p(f|\theta,\mathcal{D})q(\theta|\mathcal{D})(f(X_U)_i - \mathbb{E}_p[f(X_U)_i])^2 d\eta(f,\theta)\right).$$

Note that the regularization function only depends on $\bar{\theta}$, through $q(\theta \mid \mathcal{D}) = \delta_{\bar{\theta}}(\theta)$. Therefore the optimal post-data posterior $q$ in the regularized objective is still in the form $q^*(f, \theta|\mathcal{D}) = p(f|\theta, \mathcal{D})\delta_{\bar{\theta}}(\theta)$, and $q$ is modified by the regularization function only through $\delta_{\bar{\theta}}(\theta)$.

Thus, using the optimal post-data posterior $q^*(f, \theta|\mathcal{D}) = p(f|\theta, \mathcal{D})\delta_{\bar{\theta}}(\theta)$, the RegBayes problem is equivalent to the objective optimized by SSDKL:

$$\operatornamewithlimits{arginf}_{q(f,\theta|\mathcal{D})\in\mathcal{P}_{prob}} \mathcal{L}(q(f,\theta|\mathcal{D})) + \Omega(q(f,\theta|\mathcal{D}))$$

$$= \operatornamewithlimits{arginf}_{\bar{\theta}} -\log p(\mathbf{y}_L|\bar{\theta},X) + \alpha'\sum_{i=1}^{m}\left(\int_{f}(f(X_U)_i - \mathbb{E}_p[f(X_U)_i])^2\int_{\theta}\delta_{\bar{\theta}}(\theta)p(f|\theta,\mathcal{D})d\eta(f,\theta)\right)$$

$$= \operatornamewithlimits{arginf}_{\bar{\theta}} -\log p(\mathbf{y}_L|\bar{\theta},X) + \alpha'\sum_{i=1}^{m}\int_{f} p(f|\bar{\theta},\mathcal{D})(f(X_U)_i - \mathbb{E}_p[f(X_U)_i])^2 d\eta(f)$$

$$= \operatornamewithlimits{arginf}_{\bar{\theta}} -\log p(\mathbf{y}_L|\bar{\theta},X) + \alpha'\sum_{i=1}^{m}\mathrm{Var}_p(f((X_U)_i))$$

$$= \operatornamewithlimits{arginf}_{\bar{\theta}} L_{semisup}(\bar{\theta}).$$

$\square$

## A.2 Virtual Adversarial Training

Virtual adversarial training (VAT) is a general training mechanism which enforces *local distributional smoothness* (LDS) by optimizing the model to be less sensitive to adversarial perturbations of the input [2]. The VAT objective is to augment the marginal likelihood with an LDS objective:

$$\frac{1}{n}\sum_{i=1}^{n}\log p(\mathbf{y}_L|X_L,\bar{\theta}) + \frac{\lambda}{n}\sum_{i=1}^{n}\text{LDS}(X_i,\bar{\theta})$$

where

$$\text{LDS}(X_i,\bar{\theta}) = -\Delta_{KL}(r_{\text{v-adv}(i)}, X_i, \bar{\theta})$$

$$\Delta_{KL}(r, X_i, \bar{\theta}) = D_{KL}(p(\mathbf{y}|X_i,\bar{\theta})\|p(\mathbf{y}|X_i + r, \bar{\theta}))$$

$$r_{\text{v-adv}(i)} = \arg\max_r \{\Delta_{KL}(r, X_i, \bar{\theta}); \|r\|_2 \leq \epsilon\}$$

and $\mathbf{y}$ is the output of the model given the input $X_i$ (or perturbed input $X_i + r$) and parameters $\bar{\theta}$. Note that the LDS objective does not require labels, so that unlabeled data can be incorporated. The experiments in the original paper are for classification, although VAT is general. We use VAT for regression by choosing $p(\mathbf{y}|X_i,\bar{\theta}) = \mathcal{N}(h_{\bar{\theta}}(X_i), \sigma^2)$ where $h_{\bar{\theta}} : \mathbb{R}^d \to \mathbb{R}^H$ is a parameterized mapping (a neural network), and $\sigma$ is fixed. Optimizing the likelihood term is then equivalent to minimizing the squared error and the LDS term is the KL-divergence between the model's Gaussian distribution and a perturbed Gaussian distribution, which is also in the form of a squared difference. To calculate the adversarial perturbation $r_{\text{v-adv}(i)}$, first we take the second-order Taylor approximation at $r = 0$ of $\Delta_{KL}(r, X_i, \bar{\theta})$, assuming that $p(y|X_i, \bar{\theta})$ is twice differentiable:

$$D_{KL}(p(\mathbf{y}|X_i,\bar{\theta})\|p(\mathbf{y}|X_i + r, \bar{\theta})) \approx \frac{1}{2}r^T H_i r \tag{15}$$

where $H_i = \nabla\nabla_r D_{KL}(p(\mathbf{y}|X_i,\bar{\theta}))|_{r=0}$. Note that the first derivative is zero since $D_{KL}(p(\mathbf{y}|X_i,\bar{\theta})$ is minimized at $r = 0$. Therefore $r_{\text{v-adv}(i)}$ can be approximated with the first dominant eigenvector of the $H_i$ scaled to norm $\epsilon$. The eigenvector calculation is done via a finite-difference approximation to the power iteration method. As in [2], one step of the finite-difference approximation was used in all of our experiments.

## A.3 Training details

In our reported results, we use the standard squared exponential or radial basis function (RBF) kernel,

$$k(\mathbf{x}_i, \mathbf{x}_j) = \phi_f^2 \exp\left(-\frac{\|\mathbf{x}_i - \mathbf{x}_j\|_2^2}{2\phi_l^2}\right),$$

where $\phi_f^2$ and $\phi_l^2$ represent the signal variance and characteristic length scale. We also experimented with polynomial kernels, $k(\mathbf{x}_i, \mathbf{x}_j) = (\phi_f \mathbf{x}_i^T \mathbf{x}_j + \phi_l)^p, p \in \mathbb{Z}_+$, but found that performance generally decreased. To enforce positivity constraints on the kernel parameters and positive definiteness of the covariance, we train these parameters in the log-domain. Although the information capacity of a non-parametric model increases with the dataset size, the marginal likelihood automatically constrains model complexity without additional regularization [3]. The parametric neural networks are regularized with L2 weight decay to reduce overfitting, and models are implemented and trained in TensorFlow using the ADAM optimizer [4, 5].

## A.4 UCI results

In section 4.2 of the main text, we include results on the UCI datasets for $n = 100$ and $n = 300$. Here we include the rest of the experimental results, for $n \in \{50, 200, 400, 500\}$. For $n = 50$, we note that both COREG and label propagation perform quite well — we expect that this is true because these methods do not require learning neural network parameters from data.

| | | | Percent reduction in RMSE compared to DKL | | | | | |
| | | | $n = 50$ | | | | | |
| Dataset | $N$ | $d$ | SSDKL | CoREG | Label Prop | VAT | Mean Teacher | VAE |
|---|---|---|---|---|---|---|---|---|
| Skillcraft | 3,325 | 18 | 5.67 | **8.52** | 7.60 | 3.92 | -12.01 | -19.93 |
| Parkinsons | 5,875 | 20 | **-8.34** | -18.18 | -32.85 | -83.51 | -69.98 | -95.57 |
| Elevators | 16,599 | 18 | 4.92 | 5.83 | **11.28** | -8.19 | -20.91 | -16.35 |
| Protein | 45,730 | 9 | -0.54 | 5.05 | **7.52** | 0.22 | 5.51 | 4.57 |
| Blog | 52,397 | 280 | 7.69 | 8.66 | **8.71** | 8.40 | 6.89 | 6.26 |
| CTslice | 53,500 | 384 | -13.92 | **-2.14** | -17.83 | -36.95 | -35.45 | -33.24 |
| Buzz | 583,250 | 77 | 5.56 | **22.21** | 18.52 | 1.64 | -62.65 | -41.81 |
| Electric | 2,049,280 | 6 | **32.41** | -34.82 | -64.45 | -105.74 | -179.13 | -201.51 |
| Median | | | 5.24 | 5.44 | **7.56** | -3.99 | -28.18 | -26.59 |

Table 1: Percent reduction in RMSE compared to baseline supervised deep kernel learning (DKL) model for semi-supervised deep kernel learning (SSDKL), CoREG, label propagation, variational auto-encoder (VAE), mean teacher, and virtual adversarial training (VAT) models. Results are averaged across 10 trials for each UCI regression dataset. Here $N$ is the total number of examples, $d$ is the input feature dimension, and $n = 50$ is the number of labeled training examples. Final row shows median percent reduction in RMSE achieved by using unlabeled data.

| | | | Percent reduction in RMSE compared to DKL | | | | | |
| | | | $n = 200$ | | | | | |
| Dataset | $N$ | $d$ | SSDKL | CoREG | Label Prop | VAT | Mean Teacher | VAE |
|---|---|---|---|---|---|---|---|---|
| Skillcraft | 3,325 | 18 | 7.79 | 0.51 | **7.96** | 4.43 | -22.26 | -20.11 |
| Parkinsons | 5,875 | 20 | **1.45** | -29.86 | -48.93 | -160.51 | -132.12 | -195.88 |
| Elevators | 16,599 | 18 | **12.80** | -10.23 | -5.51 | -33.00 | -32.94 | -42.74 |
| Protein | 45,730 | 9 | **2.49** | -0.56 | 1.99 | -8.96 | -8.57 | -8.65 |
| Blog | 52,397 | 280 | 4.16 | 9.87 | **14.78** | 14.01 | 8.09 | 7.88 |
| CTslice | 53,500 | 384 | -11.96 | **-3.05** | -7.82 | -43.25 | -67.95 | -55.53 |
| Buzz | 583,250 | 77 | 4.78 | **8.60** | -2.93 | -30.94 | -106.85 | -103.69 |
| Electric | 2,049,280 | 6 | **-2.72** | -166.86 | -292.88 | -432.04 | -580.78 | -722.28 |
| Median | | | **3.32** | -1.81 | -4.22 | -31.97 | -50.45 | -49.13 |

Table 2: See Table 1 above and section 4.2 in the main text for details, results for $n = 200$ labeled examples.

## A.5 Poverty prediction

High-resolution satellite imagery offers the potential for cheap, scalable, and accurate tracking of changing socioeconomic indicators. The United Nations has set 17 Sustainable Development Goals (SDGs) for the year 2030—the first of these is the worldwide elimination of extreme poverty, but a lack of reliable data makes it difficult to distribute aid and target interventions effectively. Traditional data collection methods such as large-scale household surveys or censuses are slow and expensive, requiring years to complete and costing billions of dollars [6]. Because data on the outputs that we care about are scarce, it is difficult to train models on satellite imagery using traditional supervised methods. In this task, we attempt to predict local poverty measures from satellite images using limited amounts of poverty labels. As described in [7], the dataset consists of $3,066$ villages across five Africa countries: Nigeria, Tanzania, Uganda, Malawi, and Rwanda. These countries include some of the poorest in the world (Malawi, Rwanda) as well as regions of Africa that are relatively better off (Nigeria), making for a challenging and realistically diverse problem. The raw satellite inputs consist of $400 \times 400$ pixel RGB satellite images downloaded from Google Static Maps at zoom level 16, corresponding to $2.4$ m ground resolution. The target variable that we attempt to predict is a wealth index provided in the publicly available Demographic and Health Surveys (DHS) [8, 9].

## References

[1] Jun Zhu, Ning Chen, and Eric P Xing. Bayesian inference with posterior regularization and applications to infinite latent svms. *Journal of Machine Learning Research*, 15(1):1799–1847, 2014.

|  |  |  | Percent reduction in RMSE compared to DKL | | | | | |
|  |  |  | $n = 400$ | | | | | |
| Dataset | $N$ | $d$ | SSDKL | COREG | Label Prop | VAT | Mean Teacher | VAE |
|---|---|---|---|---|---|---|---|---|
| Skillcraft | 3,325 | 18 | -0.21 | -5.28 | **0.76** | -2.56 | -34.21 | -33.01 |
| Parkinsons | 5,875 | 20 | **7.92** | -20.65 | -75.10 | -191.56 | -154.43 | -234.07 |
| Elevators | 16,599 | 18 | **-1.19** | -38.84 | -32.25 | -72.48 | -83.24 | -72.90 |
| Protein | 45,730 | 9 | -1.57 | -0.02 | **0.35** | -12.02 | -10.90 | -11.59 |
| Blog | 52,397 | 280 | -2.47 | 4.48 | **6.05** | 5.28 | 0.60 | -0.68 |
| CTslice | 53,500 | 384 | **15.21** | 5.35 | 7.38 | -42.73 | -68.37 | -66.05 |
| Buzz | 583,250 | 77 | **3.94** | 3.37 | -9.86 | -40.47 | -118.13 | -119.55 |
| Electric | 2,049,280 | 6 | **-5.47** | -159.98 | -319.97 | -504.63 | -680.03 | -866.89 |
| Median |  |  | **-0.70** | -2.65 | -4.76 | -41.60 | -75.81 | -69.47 |

Table 3: See Table 1 above and section 4.2 in the main text for details, results for $n = 400$ labeled examples.

|  |  |  | Percent reduction in RMSE compared to DKL | | | | | |
|  |  |  | $n = 500$ | | | | | |
| Dataset | $N$ | $d$ | SSDKL | COREG | Label Prop | VAT | Mean Teacher | VAE |
|---|---|---|---|---|---|---|---|---|
| Skillcraft | 3,325 | 18 | **-5.59** | -10.35 | -6.64 | -9.11 | -31.52 | -32.09 |
| Parkinsons | 5,875 | 20 | **9.42** | -15.48 | -56.79 | -198.14 | -157.34 | -240.18 |
| Elevators | 16,599 | 18 | **0.82** | -43.95 | -39.11 | -80.17 | -93.15 | -96.91 |
| Protein | 45,730 | 9 | **-1.19** | -3.36 | -3.24 | -17.73 | -14.64 | -16.60 |
| Blog | 52,397 | 280 | 3.37 | 7.58 | **12.85** | 10.56 | 2.23 | 5.01 |
| CTslice | 53,500 | 384 | **5.80** | 3.50 | -4.35 | -73.67 | -86.25 | -115.66 |
| Buzz | 583,250 | 77 | **7.38** | 2.83 | -13.52 | -42.03 | -137.47 | -112.36 |
| Electric | 2,049,280 | 6 | **-8.71** | -136.52 | -301.95 | -472.13 | -635.63 | -836.90 |
| Median |  |  | **2.09** | -6.85 | -10.08 | -57.85 | -89.70 | -104.63 |

Table 4: See Table 1 above and section 4.2 in the main text for details, results for $n = 500$ labeled examples.

[2] Takeru Miyato, Shin-ichi Maeda, Masanori Koyama, Ken Nakae, and Shin Ishii. Distributional smoothing with virtual adversarial training. *arXiv preprint arXiv:1507.00677*, 2015.

[3] Carl Edward Rasmussen and Christopher KI Williams. *Gaussian processes for machine learning*. The MIT Press, 2006.

[4] Martın Abadi, Ashish Agarwal, Paul Barham, Eugene Brevdo, Zhifeng Chen, Craig Citro, Greg S Corrado, Andy Davis, Jeffrey Dean, Matthieu Devin, et al. Tensorflow: Large-scale machine learning on heterogeneous distributed systems. *arXiv preprint arXiv:1603.04467*, 2016.

[5] Diederik Kingma and Jimmy Ba. Adam: A method for stochastic optimization. *3rd International Conference for Learning Representations*, 2015.

[6] Morten Jerven. *Poor numbers: how we are misled by African development statistics and what to do about it*. Cornell University Press, 2013.

[7] Neal Jean, Marshall Burke, Michael Xie, W Matthew Davis, David B Lobell, and Stefano Ermon. Combining satellite imagery and machine learning to predict poverty. *Science*, 353(6301):790–794, 2016.

[8] ICF International. Demographic and health surveys (various) [datasets], 2015.

[9] Deon Filmer and Lant H Pritchett. Estimating wealth effects without expenditure data—or tears: An application to educational enrollments in states of india*. *Demography*, 38(1):115–132, 2001.