[Reviews · NeurIPS 2018]

Reviewer 1



The paper proposes a way to leverage unlabeled data for training deep kernel learning methods. The authors construct a loss function in the form of the marginal likelihood regularized by the variance of the predictive distribution at the unlabeled points and prove that minimizing their semisupervised objective is equivalent to solving a regularized posterior optimization problem. The paper is well written, easy to follow, and has a clear motivation and strong empirical results (the method compares favorably against popular baselines). A few questions/comments for the authors: - Does the proposed regularizer impact the calibration of the model? The intuition for regularizing the model using unlabeled data is clear (e.g., from the representation learning point of view), but it is unclear whether or not directly minimizing the variance on unlabeled points results into an over-confident model. It would be interesting to see how that turns out in practice. - The authors mention inductive and transductive types of semi-supervised learning in the introduction but it seems like they do experiments only in the inductive setting (if I understand correctly). Would be nice to see results for the transductive setup (or make it explicit that the focus of the paper is on inductive regime). - Would be nice if the authors can include a note on the implementation of their models and whether they plan to release code/data.

Reviewer 2



The paper paper applies Deep Kernel Learning [DKL, 1] to Semi-Supervised Regression. DKL is a combination of a Gaussian Process and a Deep Neural Network (DNN). The idea is to use DNN as a feature transformer inside the kernel of a Gaussian Process (GP). In other words, the the GP operates on the outputs of the DNN. Both the GP and the DNN can be trained using SGD in end-to-end fashion. This paper proposes to apply DKL to semi-supervised regression by minimizing the posterior predictive variance on the unlabeled data. Authors demonstrate improved performance on semi-supervised problems compared to conventional DKL (trained on labeled data only) as well as adapted versions of semi-supervised classification techniques such as VAT [2]. The paper is technically correct and well-written. I have the following questions and concerns: 1. Do I understand correctly that the idea of variance minimization is specific to DKL and would not work for GPs with conventional kernels? The variance of the predictive distribution serves as a proxy for how the unlabeled data points are close to labeled data in the output space of the DNN? While the paper spends several paragraphs on the motivation for minimizing the predictive variance I still find it somewhat unclear. In the rebuttal the authors confirmed my claim above. I encourage them to try to make the motivation more clear for the camera-ready version. 2. To the best of my knowledge this paper is the first to consider deep semi-supervised regression. Thus, there are no known reference results that could be used to compare against the proposed method. The authors adapt several SSL techniques such as VAT [2] to regression for the sake of comparison, but it is not clear if the baselines are tuned sufficiently. For this reason, it would be beneficial to evaluate the proposed technique on semi-supervised classification, where multiple baseline results are available. The authors leave semi-supervised classification for future work. To address this issue the authors will publicly release their code according to the rebuttal. 3. The authors use cross-validation for tuning hyper-parameters. Is it possible to tune the hyper-parameters by maximizing the likelihood / evidence lower bound (ELBO)? As discussed in [3], only very small validation sets can be used in realistic applications of semi-supervised learning; using the bayesian model selection can be especially beneficial in this application, if it would allow to avoid using a validation set. According to the rebuttal the authors tune some of the hyper-parameters with marginal likelihood optimization, and will make this point more clear in the camera-ready version. The paper explores semi-supervised regression, an unexplored but potentially important area of machine learning. It proposes new interesting methodology for this problem and shows promising results. I recommend an accept for this paper. I was satisfied with the author's feedback and the discussion phase didn't raise any important issues with this paper. I thus increase my score to 7. [1] Deep Kernel Learning; Andrew Gordon Wilson, Zhiting Hu, Ruslan Salakhutdinov, Eric P. Xing [2] Virtual Adversarial Training: A Regularization Method for Supervised and Semi-Supervised Learning; Takeru Miyato, Shin-ichi Maeda, Masanori Koyama, Shin Ishii [3] Realistic Evaluation of Deep Semi-Supervised Learning Algorithms; Avital Oliver, Augustus Odena, Colin Raffel, Ekin D. Cubuk, Ian J. Goodfellow

Reviewer 3



Thanks for the clarification. It'd be good to add the complexity discussion to the revised version as it's usually a big concern for GP-based methods. ---- This paper extends the deep kernel learning approach to enable semi-supervised learning. The method is to minimize the predictive variance on unlabeled data. The approach is connected to the posterior regularization and the model is learned with the variational framework. Improved results are obtained on an extensive set of datasets/tasks. The intuition of minimizing the predictive variance on unlabeled data is reasonable and well motivated in the paper. The math development of the algorithm looks sound. The experimental performance is impressive compared to baselines like DKL. A practical issue of Gaussian processes is the computational efficiency. The base DKL algorithm put much emphasis on approximate computations that reduce the computation complexity to linear wrt to data size. Did the algorithm here use the same approximations? What's the computation complexity wrt data size? What is the training time of the experiments?